# Identification of Potential Targets Linked to the Cardiovascular/Alzheimer’s Axis through Bioinformatics Approaches

**DOI:** 10.3390/biomedicines10020389

**Published:** 2022-02-06

**Authors:** Francisco Andújar-Vera, Cristina García-Fontana, Raquel Sanabria-de la Torre, Sheila González-Salvatierra, Luis Martínez-Heredia, Iván Iglesias-Baena, Manuel Muñoz-Torres, Beatriz García-Fontana

**Affiliations:** 1Instituto de Investigación Biosanitaria de Granada, 18012 Granada, Spain; raquelsanabriadlt@gmail.com (R.S.-d.l.T.); sgsalvatierra@ugr.es (S.G.-S.); luismh95@gmail.com (L.M.-H.); bgfontana@fibao.es (B.G.-F.); 2Department of Computer Science and Artificial Intelligence, University of Granada, 18071 Granada, Spain; 3Andalusian Research Institute in Data Science and Computational Intelligence (DaSCI Institute), 18014 Granada, Spain; 4Endocrinology and Nutrition Unit, University Hospital Clínico San Cecilio of Granada, 18016 Granada, Spain; 5CIBERFES, Instituto de Salud Carlos III, 28029 Madrid, Spain; 6Department of Medicine, University of Granada, 18016 Granada, Spain; 7Fundación para la Investigación Biosanitaria de Andalucía Oriental-Alejandro Otero (FIBAO), 18012 Granada, Spain; iglesiasbaena@hotmail.com

**Keywords:** Alzheimer’s disease, bioinformatics, cardiovascular disease, differentially expressed genes, hubs, protein–protein interaction network

## Abstract

The identification of common targets in Alzheimer’s disease (AD) and cardiovascular disease (CVD) in recent years makes the study of the CVD/AD axis a research topic of great interest. Besides aging, other links between CVD and AD have been described, suggesting the existence of common molecular mechanisms. Our study aimed to identify common targets in the CVD/AD axis. For this purpose, genomic data from calcified and healthy femoral artery samples were used to identify differentially expressed genes (DEGs), which were used to generate a protein–protein interaction network, where a module related to AD was identified. This module was enriched with the functionally closest proteins and analyzed using different centrality algorithms to determine the main targets in the CVD/AD axis. Validation was performed by proteomic and data mining analyses. The proteins identified with an important role in both pathologies were apolipoprotein E and haptoglobin as DEGs, with a fold change about +2 and −2, in calcified femoral artery vs healthy artery, respectively, and clusterin and alpha-2-macroglobulin as close interactors that matched in our proteomic analysis. However, further studies are needed to elucidate the specific role of these proteins, and to evaluate its function as biomarkers or therapeutic targets.

## 1. Introduction

Cardiovascular disease (CVD) is the leading cause of mortality worldwide. The World Health Organization (WHO) estimates that in 2015 (the latest year for which data have been published), 31% of all deaths worldwide were due to CVD. CVD is mainly caused by the development of atherosclerosis, and encompasses coronary heart disease, peripheral arterial disease and cerebrovascular disease. There are several well-known risk factors that increase the likelihood of developing CVD, such as hypertension, smoking, decreased serum high-density-lipoprotein (HDL), increased serum low-density lipoprotein (LDL), diabetes, sedentary lifestyle, obesity, family history, age, and alcohol consumption. Scientific evidence has shown that CVD could be related to other pathologies which a priori might appear to be independent of each other, such as bone [1,2,3] or neurological disorders [4,5]. In this context, experimental and clinical studies on brain ischemia provide ample evidence that ischemia is involved in the development of the phenotype and genotype of Alzheimer’s disease (AD) [6].

AD, defined by the WHO as a neurodegenerative disease of unknown etiology characterized by a progressive deterioration of memory and cognitive function [7], accounts for approximately 50–75% of all cases of dementia. According to worldwide statistics from the Global Burden Disease Study, AD was among the 50 main causes of mortality in the period between 1990 and 2013. There are currently about 34 million people with AD worldwide and the prevalence is expected to triple in the next 40 years due to demographic changes and longer life expectancy, which translates to 1 in 85 people worldwide being affected with AD by 2050 [8]. Risk factors that have been associated with the development of AD include female gender [9], age [10], smoking [11], obesity [12,13] and diabetes mellitus [14], among others.

Traditionally, attempts to understand AD have focused on trying to predict the presence of amyloid plaques and neurofibrillary tangles, and on understanding the origin of their accumulation. However, there is growing scientific evidence linking it to other molecular determinants, such as apolipoprotein E, lipid metabolism, neuroinflammation and mitochondrial function, that appear to play a key role in the development of AD [15].

Although aging is the most important risk factor for the development of CVD and AD [16], cardiovascular-related risk factors play a key role in cognitive disorders by their involvement in the amyloid clearance process. For instance, several cardiovascular risk factors, such as hyper and hypotension, heart failure, coronary artery disease, stroke, emboli and atherosclerosis, have been reported as potential factors with a role in cognitive decline in AD patients [17,18,19]. 

Similarly, obesity, a major CVD risk factor, has been consistently associated with the presence of dementia, as well as a higher midlife body mass index, which proportionally increases AD risk [20].

On the other hand, brain alterations in AD patients are often accompanied by vascular alterations, including blood vessels with collapsed or degenerated endothelia in more than 90% of cases [21]. Other studies have reported the presence of amyloid deposits in distal organs, vessels and the heart of AD patients, causing damage to these organs and promoting the development of cardiovascular complications, especially heart diastolic dysfunction [18,19,22]. In addition, CVD patients and AD patients share common brain structural alterations [22]. 

Scientific evidences from genetic studies have also supported the link between CVD and AD [23]. The study of Ray et al. (2008), through a computational analysis, shows the co-existence in a single functional module of genes typically related to AD and genes associated with CVD. This study shows the existence of extensive links between AD and CVD in terms of co-expression and co-regulation [24]. Moreover, genetic meta-analyses of AD have identified new risk loci involved in lipid processing highly related to CVD [25]. 

In the therapeutic context, there is some scientific evidence which supports this link between CVD and AD. Thus, a beneficial effect has been observed in the use of certain direct oral anticoagulants, such as dabigatran, rivaroxaban and apixaban, for the treatment of cerebral amyloid angiopathy. These molecules have therefore been proposed to improve the vascular-mediated progression of neurodegenerative and cognitive changes in AD [26]. 

In this line, the evidence supports that the short- [25] and long-term [26] administration of dabigatran can induce a recovery of cognitive impairment, as well as decrease oxidative stress and neuroinflammation, and reduce the deposition of amyloid plaques [27]. 

Consistent with these findings, a recent article using computational tools of virtual screening and molecular docking simulations describes the use of potential multimodal agents, such as certain coagulation factors (thr and/or fXa), for the treatment of AD [26].

Furthermore, epidemiological studies have also supported the coexistence of both pathologies, showing a positive correlation between dementia and CVD in several populations. For instance, both disorders have been reported in 18% of a dementia autopsy series [27] and in individuals with congestive heart failure [28]. Cerebral ischemia and stroke may lead to hypoxia, amyloid beta (Aβ) deposition, and impairment of the blood–brain barrier, leading to neuronal degeneration [29]. Therefore, we hypothesize that there are overlapping molecular mechanisms in CVD and AD.

Considering the high proportion of patients affected by CVD, it would be of great interest to identify those with a higher risk of developing cognitive deterioration in order to establish early preventive and therapeutic measures, and to delay the onset of neurological disorders. On the other hand, it could also be useful to study the potential development of cardiovascular complications in patients diagnosed with AD in order to prevent irreversible damages. Hence, we aim to characterize common factors in the development of CVD and AD by using bioinformatics and experimental approaches to generate biological networks to identify the essential proteins (hubs) that play a key role in the signaling and regulatory processes [30] in both pathologies. The identification of these proteins could be useful to study their potential usefulness as early biomarkers of AD in patients with CVD.

## 2. Materials and Methods

### 2.1. Data Acquisition

The first step in acquiring the data of interest was to search the GEO database (https://www.ncbi.nlm.nih.gov/geo/; accessed on 19 November 2021) based on the terms “artery”, “femoral artery” or “artery atherosclerosis”. The requirements for selecting the datasets included the following aspects: (a) tissue samples from human femoral arteries; (b) in arteries without atherosclerotic lesions, the source had to be from healthy organ donors; and (c) at least 10 samples per group.

### 2.2. Data Preprocessing and Identification of Differentially Expressed Genes (DEGs)

The DEGs between calcified and healthy artery samples from patients aged 67 ± 9 years (85% males) were analyzed using GEO2R (https://www.ncbi.nlm.nih.gov/geo/geo2r/) (MD, USA), allowing the comparison of two or more datasets in a GEO series [31]. The adjusted *p*-values (adj. p) and Benjamini and Hochberg false discovery rate were applied to provide a balance between the discovery of statistically significant genes and limitations of false-positives. The absolute value of log fold change (FC) ≥ 1.5 and adj. *p* < 0.05 were considered as statistically significant.

### 2.3. Protein–Protein Interaction (PPI) Network Performance and Module Analysis

The PPI network was predicted using the Search Tool for the Retrieval of Interacting Genes/Proteins (STRING, v. 1.7.0) application [32] in the Cytoscape program (v. 3.9.0) [33]. Cytoscape is an open-source bioinformatic software platform for visualizing molecular interaction networks. The proteins that were used as seeds for the creation of the PPI network were the DEGs obtained in the previous section. For the initial exploration, the confidence score was set to high (score 0.007). Proteins without matches in the STRING database and proteins without interactions with other proteins within the network were discarded. 

To find the highest connected regions based on topology within the whole PPI network, the Molecular Complex Detection (MCODE, v. 2.0.0) [34], a Cytoscape plug-in, was used. The criteria for selection were as follows: degree cutoff = 2, K-core = 2, node score cutoff = 0.2 and max depth up to 100. Thus, highly interconnected regions (modules) in the network were identified and, therefore, the functionality of the modules was studied in depth.

Subsequently, the analyses of genes associated with diseases in these modules and the Gene Ontology (GO) enrichment analysis were performed using the DAVID online database (https://david.ncifcrf.gov/home.jsp) (MD, USA) to identify the modules of interest related to AD.

### 2.4. Construction of a Secondary Related PPI Network and Identification of Hubs

The proteins belonging to the module of interest identified in the previous section were selected for the construction of a new PPI network. Similar to the prediction of the first PPI network, Cytoscape’s STRING application was used. This secondary PPI network kept the same high confidence score of 0.007. In addition, the maximum number of interactors to display was set in 50.

Based on the open-source platform Cytoscape, a convenient app called CytoNCA (v.2.1.6) [35] for network centrality analysis was used to explore the secondary PPI network. The two algorithms used were Degree and Betweenness, to identify important nodes in a large number of interactions and the total number of shortest pathways between two nodes, respectively. Through this tool, the most important nodes within the network (called hubs) were identified. Additionally, a KEGG and GO enrichment analysis was performed in the DAVID database on the secondary PPI network. This analysis allowed for the identification of the most significant GO terms and pathways for this set of proteins.

### 2.5. Validation of Target Proteins by Data-Mining and Proteomic Analysis

Factors associated with AD were identified using DisGeNET (http://www.DisGeNET.org/home/) (Spain) [36], a discovery platform which integrates information on GDAs from several public data sources and the scientific literature about gene expression, biomarkers, clinical phenotype associations with the corresponding diseases, variant-disease associations, and single nucleotide polymorphisms. The current version of, DisGeNET (v.7.0) (Spain) contains 1,134,942 GDAs between 21,671 genes and 30,170 diseases and traits.

A proteomic analysis of calcified femoral artery samples from 7 patients diagnosed with type 2 diabetes, according to the criteria established in the American Diabetes Association (2011), was performed to validate the results obtained through bioinformatic approaches. The cohort of patients was represented by adult males (mean age 74 ± 10 years) who were monitored at the University Hospital Clínico San Cecilio of Granada. All patients were diagnosed with critical ischemia, according to the consensus document on peripheral disease TASC II [37], with an indication of lower limb amputation. Briefly, protein samples from the vascular tissue were extracted, followed by concentration, clean up and digestion standard procedures. Then, protein separation and identification were performed using nano-scale liquid chromatographic tandem mass spectrometry (nLC-MS/MS) and Proteome Discoverer, respectively, as previously reported [38] 

The detailed protocol is extensively described in the Appendix A. 

Both the disease-associated indicators obtained from DisGeNET and the proteins identified in the proteomic analysis were confronted with the list of proteins of interest obtained in the PPI network.

The workflow schematic performed in this study is summarized in Figure 1.

## 3. Results

### 3.1. Data Acquisition

After an exhaustive search, the microarray expression profile dataset GSE100927 were downloaded from the GEO database, which was based on GPL17077 Agilent-039494 SurePrint G3 Human GE v2 8x60K Microarray 039381. The GSE100927 dataset included 26 atheromatous plaque samples harvested from patients undergoing femoral endarterectomy and 12 healthy artery samples free of atherosclerotic lesions obtained from organ donors [39]. 

### 3.2. Data Preprocessing and Identification of DEGs

The selection of DEGs analyzed with GEO2R between the femoral artery with atherosclerotic lesions and the healthy femoral arteries are presented in a volcano diagram (Figure 2). A total of 163 DEGs were identified in atheromatous plaques compared to controls, including 143 upregulated and 20 downregulated DEGs (Appendix A).

### 3.3. PPI Network Performance and Module Analysis

The 163 DEGs obtained in the previous section were entered into the STRING application. Once the filters were set for the whole dataset, a PPI network including 79 proteins (nodes) and 166 interactions (edges) resulted (Figure 3A).

The analysis performed by MCODE for the determination of the network modules resulted in a prediction of six modules including highly connected nodes (Figure 3B).

Each of the modules obtained was analyzed through the DAVID enrichment tool, and only the module most related to AD was selected (module 2, Figure 3B). Table 1 shows the enrichment results of module 2 obtained after analysis by the DAVID tool. The other modules did not offer obvious linkage with the pathologies under study.

### 3.4. Enrichment of the Secondary PPI Network and Identification of Hubs

Once the AD-related module was identified, the proteins forming the module were selected for visualization (Figure 4A) and enriched with the 50 functionally nearest proteins (Figure 4B). 

For the determination of the most important proteins (hubs) from the secondary PPI network, CytoNCA was used by applying the “Betweenness” and “Degree” centrality algorithms. Table 2 shows the result of the 15 proteins with the highest centrality value within this secondary PPI network derived from the initial module 2.

GO and KEGG enrichment analysis shows that most genes of the secondary PPI network are involved in pathways related to lipid metabolism, and inflammation processes, as well as immune response. Some of them also appear to be involved in AD-related pathways (Appendix A).

### 3.5. Validation of Target Proteins by Data-Mining and Proteomic Analysis

Both datamining-based techniques and laboratory proteomic analysis techniques were used for the validation.

The genes associated with AD were obtained through the DisGeNET platform. Once the cutoff threshold was set at 0.3, the gene–disease associations (GDAs) were included in the Venn diagram (Figure 5A). A total of 124 indicators of AD were identified in this step.

On the other hand, proteomic analysis provided a total of 751 proteins, which were added to the Venn diagram. After cleaning the data by removing duplicate, empty or unmatched entries from the Uniprot database, 737 total proteins resulted (Figure 5B).

Both previous sets were confronted with the proteins constituting the AD-related secondary PPI network derived from the initial module 2 containing a total of 55 proteins (Figure 5C).

## 4. Discussion

The links between neurodegenerative disorders such as AD and CVD have not currently been elucidated. However, studies supporting this connection are gradually appearing [22]. Frequently, vascular changes accompany or even precede the development of AD, raising the possibility that they may have a pathogenic role. The fact that aging is the most important risk factor for the development of CVD and AD [16], and that common brain structural alterations have been detected in both pathologies, suggest that there is an overlap in the molecular mechanisms shared by CVD and AD [22]. 

The results of this study show, through PPI network analysis and enrichment analysis, a set of genes associated with the presence of both CVD and AD. Specifically, we found three targets (APOE, CLU and A2M) closely related to both pathologies. In addition, other proteins such as haptoglobin (HP) should be considered in this context since, although we found controversial evidence, considering the consulted databases in our bioinformatics study we found strong evidence in the scientific literature that could link this target to the CVD/AD axis.

It does not seem to be a coincidence that there is a relationship between CVD and AD, since the role of lipids in the development of atherosclerosis is well known [40], and several studies have shown the key involvement of proinflammatory lipids in the formation of amyloid plaque and the aggravation of AD. Accordingly, our GO and KEGG results strongly suggest that lipid metabolism and inflammation are key factors in CVD and AD. In this context, large LDL particles have been found to correlate significantly with increased brain amyloidosis and decreased hippocampal volume [41]. Likewise, the study by Chatterjee et al. (2016) shows impaired phospholipid and sphingolipid metabolism in patients with familial forms of AD, linking the presence of these lipids with cerebrospinal fluid amyloid and tau protein [42]. On the other hand, apolipoprotein A-1 deficiency has been observed to be associated with excessive cholesterol accumulation and increased cortical exposure to amyloids, whereas apolipoprotein A-I decreases Aβ aggregation and toxicity [43]. Ceramides are other lipidic molecules that have revealed their involvement in the pathogenesis of AD, showing a clear association between plasma ceramide levels and neuropsychiatric symptoms in AD patients [44]. The study by Kim et al. (2018) revealed an impaired lipid profile in AD patients [45] and identified lipid signatures able to predict AD progression and brain atrophy [46]. Although all these findings point to a direct relationship between lipid profile with CVD and AD, clear efficacy of statins for the treatment of AD could not be established [47].

On the other hand, apolipoprotein E (APOE) is a well-known factor associated with both the development of CVD [48,49,50] and the development of AD [51,52,53,54,55,56]. However, there is little evidence of the involvement of this protein in the CVD/AD axis jointly [57].

In our study, we identified APOE as a key protein in the development of CVD and AD. Our results showed a strong involvement of APOE in the connection of both pathologies. GEO2R statistical analysis of previously published differential expression data between calcified femoral artery and non-calcified femoral artery data published by Steenman et al. (2018) showed that APOE is upregulated (two-fold) in calcified tissue. Likewise, at the qualitative level, we identified this protein in the proteome of calcified femoral arteries in patients with lower limb amputation. At the AD level, our analyses in DisGeNET and PPI network studies indicate a central role of APOE in the development of this pathology. The score obtained in the centrality algorithms places it as one of the most important hubs in the CVD/AD axis, supporting the important dual role of this protein.

In addition to APOE, we have been able to identify other targets that, although they show a somewhat lower score, can be classified as essential proteins in the development of CVD and AD. In this block, we can include alpha-2-macroglobulin (A2M), clusterin (CLU) or apolipoprotein J, and haptoglobin (HP).

A2M acts as an antiprotease and is able to inactivate an enormous variety of proteinases. It acts by inhibiting plasmin and thrombin, and is therefore an inhibitor of fibrinolysis and coagulation, respectively. A2M binds numerous growth factors and cytokines, and thus acts as a transporter protein [58]. In addition, A2M seems to participate in inflammatory reactions and appears to be involved in AD. The study by Varma et al. (2017) showed a significant association of A2M in blood with markers of neuronal injury, tau and phosphorylated tau in cerebrospinal fluid, and a higher serum A2M concentration was associated with an almost three-fold increased risk of AD progression in men [59]. These data suggest that A2M could be considered a biomarker of preclinical AD, reflecting early neuronal injury [59]. At the cardiovascular level, this protein seems to be involved in the development of atherosclerosis and cardiac hypertrophy, increasing the risk of developing CVD [60]. In this context, the study by Nezu et al. (2013), revealed that an increase in serum A2M levels could be involved in the pathophysiology of acute ischemic stroke [61], which would support the role of this protein in the development of CVD.

In our results, we identified A2M in the enrichment of the initial PPI network as an essential protein with several interactions with important components in the CVD/AD axis. Furthermore, at the qualitative level, we identified A2M in the proteome of calcified vascular tissue samples, so these data support the dual role of this protein in CVD and AD.

CLU, also known as apolipoprotein J, is a molecular chaperone associated with cellular debris removal and apoptosis [62]. Through this function, CLU is implicated in many oxidative stress-related diseases, including neurodegenerative diseases [63], cancer [64], inflammatory diseases [65] and aging [66,67].

In the case of neurodegenerative diseases, we can highlight the role of CLU in the development of AD [68,69]. Together with APOE, CLU is the most expressed apolipoprotein in the central nervous system [70,71], presenting important analogies with APOE such as its presence in amyloid plaques [72,73] and its capacity to bind Aβ [74,75]. Another analogy between APOE and CLU is at the functional level, such that several studies have proposed the participation of both in the elimination of Aβ from the brain [76,77]. In this context, it has been proposed that CLU could be considered as a therapeutic strategy to slow the progression of AD [78]. The study by Schrijvers et al. published in JAMA in 2011 showed a significant association of plasma CLU levels with AD prevalence and severity, but not with AD incidence [79]. The meta-analysis by Yang et al. (2018) agreed with this evidence, showing that a high concentration of CLU in the plasma and brain is associated with dementia, especially in AD [80], proposing this protein as a biomarker of cognitive impairment severity [81].

Although the role of CLU in cancer or neurological diseases has been intensively studied for three decades, the physiological functions of CLU in the context of the cardiovascular system are not as well studied. In this regard, there is some scientific evidence addressing these aspects. The study by Bradley et al. (Diabetes Care, 2019), showed increased expression of CLU at the extracellular matrix (ECM) of adipose tissue in obese versus lean women through microarray studies [82]. Recently, an important role of different adipose tissue ECM components in the development of obesity-related cardiomatabolic diseases has been observed [83,84]. In this regard, the authors of this work proposed adipocyte-derived CLU as a novel ECM-related protein linking cardiometabolic diseases and obesity through its actions in the liver. Along these lines, the study by Won et al. (2014) proposed circulating CLU as a surrogate marker of obesity-associated systemic inflammation [85]. Since CLU is induced in vascular smooth muscle cells (VSMCs) during atherosclerosis and injury-induced neointimal hyperplasia, the study by Kim et al. (2009), proposed the role of this protein as a protective strategy against the development of neointimal hyperplasia, rather than a causal response [86]. CLU is a ubiquitous protein, synthesized by numerous tissues and organs and with varied receptors including the HDL-cholesterol receptor, the low-density lipoprotein-related protein 2 (LRP/megalin), ApoER2, and the very low-density lipoprotein receptor (VLDLR), many of which are critical for cardiovascular health, and thus have recently been linked to cardiovascular and cerebrovascular effects [87,88,89].

Our results show the presence of CLU in the proteome of calcified vascular tissue at the qualitative level, as well as the involvement of this protein in the secondary AD-related PPI network. Our enrichment studies place it as an essential hub due to its involvement in CVD and AD. However, the controversy generated around the role of CLU makes it necessary to perform functional studies to understand the real role of CLU, and determine whether it plays a beneficial or detrimental role in the development of CVD and AD, in order to consider this protein as a potential biomarker of CVD/AD or as a therapeutic target.

Another target identified in the present study was HP, a plasma protein synthesized in the liver whose function is to bind to free hemoglobin, forming hemoglobin–haptoglobin complexes. Increased levels of HP have been observed in inflammatory processes, and it is considered an acute phase protein. In addition, HP is involved in the innate and acquired immune systems, playing a regulatory role in various stages of cellular and humoral immunity and in the release of cytokines [90]. This protein has recently been linked to neurodegenerative disorders such as AD and mild cognitive impairment. In this line, there are only two studies relating HP levels to AD severity by showing significantly higher serum levels of HP in AD, and mild cognitive impairment compared to healthy subjects [91,92]. To the best of our knowledge, there is no link between HP and CVD. However, our results using GEO2R suggest a two-fold downward regulation of HP in the calcified femoral artery compared to the non-calcified femoral artery. Furthermore, our PPI network and proteomic analyses suggest HD protein as a target in atherosclerosis and AD. These findings place HP as an important hub within the CVD/AD axis.

The scarce scientific evidence of HP in relation to AD, together with the non-existence of studies that functionally link it to the presence of CVD, makes HP an attractive candidate for the development of studies to elucidate the involvement of this protein in the CVD/AD axis. In this context, it is necessary to deepen our knowledge of these potential targets in order to broaden the range of potential biomarkers of AD and CVD for therapeutic or early diagnostic purposes.

Our study has some limitations, including the exclusion of potential targets involved in the CVD/AD axis in the bioinformatics analysis, due to establishment of a specific cut-off point. However, to overcome this limitation, an enrichment analysis was performed to avoid the loss of potential candidates. Including biological samples from patients with AD in future experimental work will corroborate the role of the identified proteins at the neurological level, confirming the dual participation of these proteins in the CVD/AD axis. The strength of our study lies in the generation of a strategy capable of combining and jointly exploiting the information available through different methodological approaches, generating very valuable and scientifically supported leads for the identification of new potential targets, some of which have been little explored to date, opening the door to perform further translational studies assessing the role of these proteins as biomarkers and/or therapeutic targets.

In summary, our findings suggest that the atherosclerotic processes leading to CVD could also be involved in the development of neurological disorders such as AD. The high incidence of CVD and its link with AD makes it necessary to search for new diagnostic strategies to identify high-risk patients in subclinical stages. Our study identified some potential novel targets in the CVD/AD axis, including APOE, HP, CLU and A2M, being the two first proteins up- and downregulated, respectively, in atherosclerotic vascular tissue compared with healthy vascular tissue. Although future studies are needed to confirm the dual functions of these proteins in the CVD/AD axis, this study provides valuable information for the study of the usefulness of these proteins as potential early biomarkers. This would affect future predictions of predisposition to AD in patients with CVD and vice versa, and facilitate the implementation of preventive and therapeutic strategies that alleviate the most aggressive effects of these disorders, and improve the quality of life of affected patients.

## 5. Conclusions

The evidence suggests that vascular pathology is a likely pathogenic contributor to age-related dementia, including AD, and is inextricably linked to disease onset and progression. In this context, our results indicated four main targets with strong scientific evidence of involvement in the CVD/AD axis: APOE, CLU, A2M and HP. Consequently, the contribution of CVD factors highly related to neurological disorders should be considered in preventive, diagnostic, and therapeutic approaches to address one of the major health challenges of our time. 

## Figures and Tables

**Figure 1 biomedicines-10-00389-f001:**
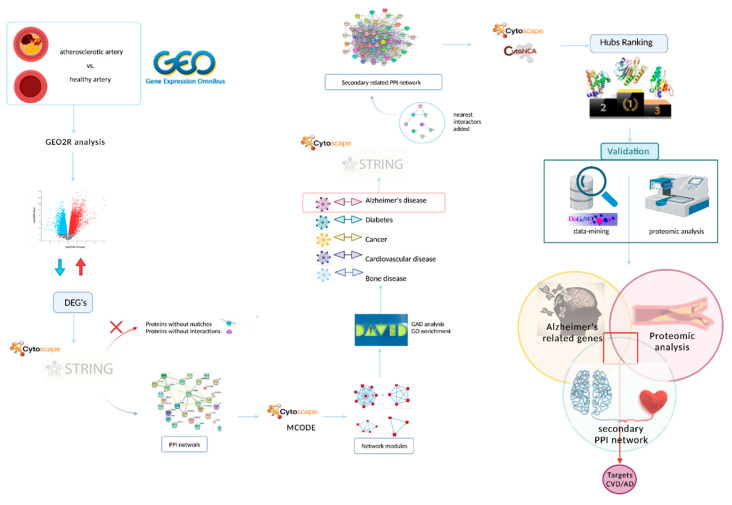
Flowchart of the methodology used. This scheme summarizes the whole process, from the search in databases of CVD and AD related targets to the validation by data-mining and proteomic analysis. DEG’s: differential expressed genes; PPI: protein–protein interaction; CVD: cardiovascular disease; AD: Alzheimer’s disease. Created with BioRender.com.

**Figure 2 biomedicines-10-00389-f002:**
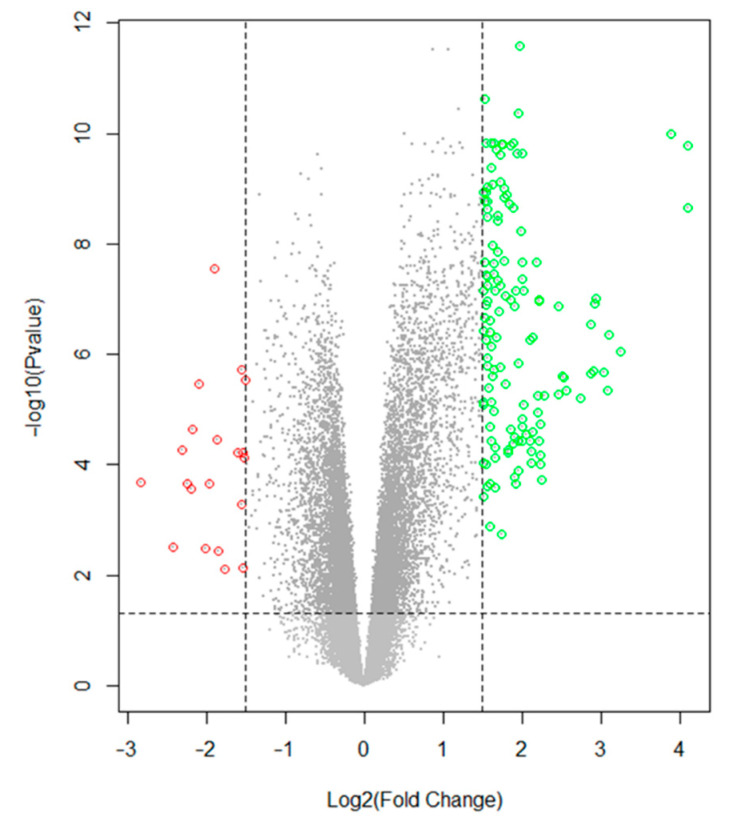
Volcano plot of differentially expressed genes (DEGs) between calcified femoral artery samples and control samples. The data were obtained from dataset GSE100927 from the GEO database. From left to right: downregulated genes (red dots), non-significant genes (grey dots, bottom panel), significant genes that do not meet the established fold change value (dark gray dots, upper panel) and upregulated genes (green dots). Log fold change ≥ 1.5 and adjusted *p*-value < 0.05 were considered as the cutoff for statistical significance.

**Figure 3 biomedicines-10-00389-f003:**
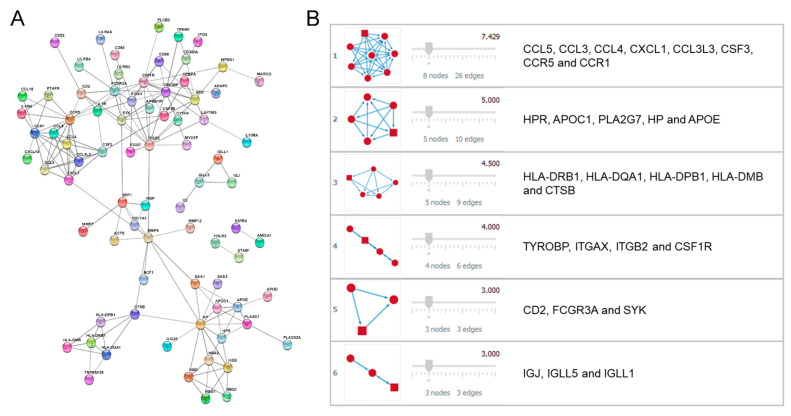
(**A**) Protein–protein interaction (PPI) network generated from differentially expressed genes (DEGs) obtained through the STRING application. (**B**) Network modules predicted through MCODE software.

**Figure 4 biomedicines-10-00389-f004:**
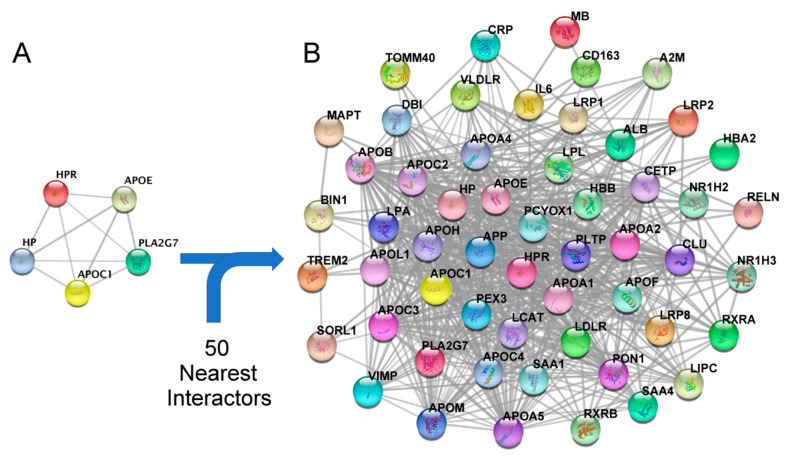
(**A**) Proteins belonging to module 2 (APOC1, HP, HPR, APOE and PLA2G7). (**B**) Protein–protein interaction (PPI) network adding the 50 functionally nearest proteins to module 2.

**Figure 5 biomedicines-10-00389-f005:**
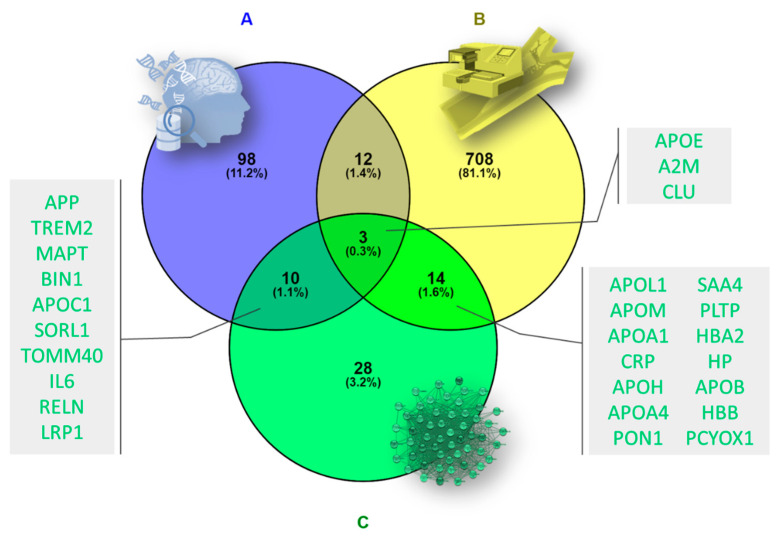
Venn diagram showing (**A**) the set of AD-related proteins obtained from DisGeNET, (**B**) the set of proteins identified in the proteome of calcified femoral arteries samples, and (**C**) the set of proteins belonging to the secondary protein–protein interaction (PPI) network derived from the initial AD-related module 2. APP: amyloid precursor protein; TREM2: triggering receptor expressed on myeloid cells 2; MAPT: microtubule-associated protein tau; BIN1: bridging integrator 1; APOC1: apolipoprotein C1; SORL1: sortilin-related receptor 1; TOMM40: translocase of outer mitochondrial membrane 40; IL6: interleukin 6; RELN: reelin; LRP1: LDL receptor-related protein 1; APOE: apolipoprotein E; A2M: alpha 2-macroglobulin; CLU: clusterin; APOL1: apolipoprotein L1; APOM: apolipoprotein M; APOA1: apolipoprotein A1; CRP: C-reactive protein; APOH: apolipoprotein H; APOA4: apolipoprotein A4; PON1: paraoxonase; SAA4: serum amyloid A4; PLTP: phospholipid transfer protein; HBA2: hemoglobin subunit alpha 2; HP: haptoglobin; APOB: apolipoprotein B; HBB: hemoglobin; PCYOX1: penylcysteine oxidase 1.

**Table 1 biomedicines-10-00389-t001:** Enrichment results of the module most related to Alzheimer’s disease (Module 2).

Category	Term	%	*p*-Value	Genes
GAD_DISEASE	cardiovascular disease	80	3.56 × 10^−7^	APOC1, HP, APOE, PLA2G7
GAD_DISEASE	atherosclerosis, coronary	80	6.64 × 10^−6^	APOC1, HP, APOE, PLA2G7
GOTERM_CC_DIRECT	GO:0005576~extracellular region	100	6.07 × 10^−5^	APOC1, HP, HPR, APOE, PLA2G7
GAD_DISEASE	cholesterol	80	1.38 × 10^−4^	APOC1, HPR, APOE, PLA2G7
GAD_DISEASE	coronary disease; coronary heart disease	60	1.61 × 10^−4^	APOC1, APOE, PLA2G7
GOTERM_CC_DIRECT	GO:0072562~blood microparticle	60	4.10 × 10^−4^	HP, HPR, APOE
GAD_DISEASE	familial dysbetalipoproteinemia	40	6.17 × 10^−4^	APOC1, APOE
GOTERM_BP_DIRECT	GO:0034447~very-low-density lipoprotein particle clearance	40	7.14 × 10^−4^	APOC1, APOE
GOTERM_BP_DIRECT	GO:0006898~receptor-mediated endocytosis	60	7.22 × 10^−4^	HP, HPR, APOE
GAD_DISEASE	type 2 diabetes; edema; rosiglitazone	100	8.23 × 10^−4^	APOC1, HP, HPR, APOE, PLA2G7
GAD_DISEASE	coronary disease; diabetes complications; hypercholesterolemia; hypertension; myocardial infarction	40	9.25 × 10^−4^	APOC1, APOE
GOTERM_MF_DIRECT	GO:0030492~hemoglobin binding	40	9.48 × 10^−4^	HP, HPR
GAD_DISEASE	cholesterol; coronary heart disease; lipoproteins	40	1.23 × 10^−3^	APOC1, APOE
GAD_DISEASE	cardiovascular diseases	60	1.24 × 10^−3^	APOC1, APOE, PLA2G7
GOTERM_MF_DIRECT	GO:0060228~phosphatidylcholine-sterol O-acyltransferase activator activity	40	1.42 × 10^−3^	APOC1, APOE
GOTERM_BP_DIRECT	GO:0034382~chylomicron remnant clearance	40	1.43 × 10^−3^	APOC1, APOE
GAD_DISEASE	Alzheimer’s disease	80	1.48 × 10^−3^	APOC1, HP, APOE, PLA2G7
GAD_DISEASE	memory disturbance	40	1.85 × 10^−3^	APOC1, APOE

Category indicates the classification shown by the DAVID database; % indicates the proportion of proteins of the module involved in the corresponding category; *p*-value consists of the modified Fisher exact *p*-value for the enrichment performed; APOC1: apolipoprotein C1; HP: haptoglobin; APOE: apolipoprotein E; PLA2G7: platelet-activating factor acetylhydrolase; HPR: haptoglobin related-protein.

**Table 2 biomedicines-10-00389-t002:** List of hubs of the secondary PPI network according to “Betweenness” and “Degree” algorithms from CytoNCA.

Symbol	Description	Degree	Betweenness
APOE	Apolipoprotein E	50.0	416.64447
APOA1	Apolipoprotein A1	42.0	118.15003
APOC2	Apolipoprotein C2	40.0	71.67373
APP	Amyloid Beta Precursor Protein	39.0	196.27913
APOA2	Apolipoprotein A2	39.0	57.75081
APOC1	Apolipoprotein C1	39.0	97.92878
APOB	Apolipoprotein B	38.0	47.32024
CLU	Clusterin	37.0	84.87135
APOC3	Apolipoprotein C3	37.0	36.22977
PLTP	Phospholipid Transfer Protein	36.0	41.08785
CETP	Cholesteryl Ester Transfer Protein	36.0	43.38615
HP	Haptoglobin	36.0	234.20578
APOA4	Apolipoprotein A4	35.0	25.99814
APOC4	Apolipoprotein C4	33.0	32.60615
APOA5	Apolipoprotein A5	33.0	416.64447

## Data Availability

The data corresponding to microarray expression profile dataset GSE100927, presented in this study are openly available in https://www.ncbi.nlm.nih.gov/geo/query/acc.cgi?acc=GSE100927, submitted by Steenman et al. (2018) [39] (doi:10.1038/s41598-018-22292-y.). The proteomic results obtained for validation are available on request from the corresponding author.

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
