# Peer review of "Identification of Potential Targets Linked to the Cardiovascular/Alzheimer’s Axis through Bioinformatics Approaches"

_biomedicines, 2022, doi:10.3390/biomedicines10020389_

Round 1

Reviewer 1 Report

The study described in the manuscript suggests several genomic pieces of evidence to support the link between cardiovascular disease (CVD) and Alzheimer's disease (AD) through bioinformatic approaches. Although the intensity of evidence provided in the manuscript is not so supportive, in my opinion, this study presents important and interesting findings that support the conventional hypothesis of an association between CVD and AD. 

The followings are my specific comments and questions: 

1. Throughout the manuscript, there are many grammatic errors and mis-spellings in English. The authors might consider consulting a language editing service. 

2. Is there a specific reason to select seven diabetic patients for a proteomic analysis?

Author Response

First, we want to thank for your effort in reviewing our manuscript and for your constructive comments, which have undoubtedly contributed to improve the quality of our manuscript. Please, find below the responses to your kind suggestions:

General comment: The study described in the manuscript suggests several genomic pieces of evidence to support the link between cardiovascular disease (CVD) and Alzheimer's disease (AD) through bioinformatic approaches. Although the intensity of evidence provided in the manuscript is not so supportive, in my opinion, this study presents important and interesting findings that support the conventional hypothesis of an association between CVD and AD. 

We are very grateful for your feedback on our work.

The followings are my specific comments and questions: 

  1. Throughout the manuscript, there are many grammatic errors and mis-spellings in English. The authors might consider consulting a language editing service. 

We appreciate your comment, based on which, we have sent our manuscript for review by the English editing service of MDPI as well as by a bi-lingual researcher with expertise in this field to improve the quality of the manuscript. We have included in the manuscript an acknowledgement section citing the collaboration of both of them in the English editing. 

  1. Is there a specific reason to select seven diabetic patients for a proteomic analysis?

Our work is carried out in the Endocrinology Unit of the Hospital Universitario San Cecilio, in collaboration with the Vascular Surgery and Angiology Unit. In this sense, we work mainly with diabetic patients, some of them with cardiovascular complications of different severity. Of this population, a small number of patients require lower limb amputation surgery. This population is our target population for the identification at the tissue level of proteins related to the development of these cardiovascular complications. The total N of our study (7 patients), reflects the patients included in the study for validation of the bioinformatic results obtained in our study.

Reviewer 2 Report

Dear authors,

The paper has been well written. The methodological approach is rational, leading to data consolidated. Discussion is very interesting, and APOE, CLU and A2M are proposed as targets closely related to both pathologies, with aim to generate new active compounds. However, I suggest to improve introduction, thus focusing well the field of investigations.

The link between cardiovascular diseases and neurodegeneration actually represents a new topics in medicinal chemistry, in the research/identification of compounds endowed of disease modifying property against AD.

Recently, some papers described as short and middle time administration of the well known antithombotic drug dabigatran etexilate can induce a recovery of cognitive impairment, as well as a reduction of amyloid plaques deposition:

  1. “Long-Term Dabigatran Treatment Delays Alzheimer's Disease Pathogenesis in the TgCRND8 Mouse Model.” Cortes-Canteli M, et al. J Am Coll Cardiol. 2019, 74, 1910-1923.
  2. “Short-term treatment with dabigatran alters protein expression patterns in a late-stage tau-based Alzheimer's disease mouse model.” Iannucci J, et al., Biochem Biophys Rep. 2020, 24,100862.
  3. “Alzheimer's Disease-Rationales for Potential Treatment with the Thrombin Inhibitor Dabigatran.” Grossmann K., Int J Mol Sci. 2021, 22, 4805).

Moreover, another recently published paper, combined inhibition of cholinesterases and factors of coagulative cascade, in a multi target directed ligands approach, thus proposing new compounds that can affect both AChE/thrombin and BChE/fXa:

  1. “First-in-Class Isonipecotamide-Based Thrombin and Cholinesterase Dual Inhibitors with Potential for Alzheimer Disease.” Purgatorio R, et.al., Molecules. 2021, 26, 5208.)

I strongly recommend citing them in introduction or in discussion.

Author Response

First, we want to thank for your effort in reviewing our manuscript and for your constructive comments, which have undoubtedly contributed to improve the quality of our manuscript. Please, find below the responses to your kind suggestions:

As you suggest, we have included information related to the work you indicate in the Introduction section.
We believe that the paragraph included (marked in red), has greatly enriched our work, since in the initial version we did not contemplate the CVD/AD link in the context of therapeutic approaches and these new references support and reinforce all the results of our study, for which we greatly appreciate the constructive contribution of your review.